



# Aerosol data assimilation in the chemical transport model MOCAGE during the TRAQA/ChArMEx campaign: Lidar observations

Laaziz El Amraoui[1], Bojan Sič[1,2], Andrea Piacentini[3], Virginie Marécal[1], Jean-Luc Attié[1,4], and Nicolas Frebourg[1]

[1]CNRM, Université de Toulouse, Météo-France, CNRS, Toulouse, France
[2]Now: at Noveltis, Toulouse, France
[3]CERFACS, Toulouse, France
[4]Also at University Paul Sabatier, Toulouse, France

**Correspondence:** Laaziz El Amraoui (laaziz.elamraoui@meteo.fr)

**Abstract.**

This paper presents the first results about the assimilation of CALIOP (Cloud-Aerosol Lidar with Orthogonal Polarization) extinction coefficient measurements on-board the CALIPSO (Cloud-Aerosol Lidar and Infrared Pathfinder Satellite Observations) satellite in the chemistry transport model MOCAGE (*Modèle de Chimie Atmosphérique à Grande Echelle*) of Météo-

France. This assimilation module is an extension of the Aerosol Optical Depth (AOD) assimilation system already presented by Sič et al. (2016). We focus on the period of TRAQA (*TRAnsport à longue distance et Qualité de l'Air dans le bassin méditerranéen*) field campaign that took place during the summer 2012. This period offers the opportunity to have access to a large set of aerosol observations from instrumented aircraft, balloons, satellite and ground-based stations. We evaluate the added value of CALIOP assimilation with respect to the model free run by comparing both fields to independent observations

issued from the TRAQA field campaign.

In this study we focus on the desert dust outbreak which happened during late June 2012 over the Mediterranean Basin (MB) during TRAQA campaign. The comparison with the AERONET (Aerosol Robotic Network) AOD measurements shows that the assimilation of CALIOP lidar observations improves the statistics compared to the model free run. The correlation between AERONET and the model (assimilation) is 0.682 (0.753), the bias and the RMSE, due to CALIOP assimilation, are reduced

from -0.063 to 0.048 and from 0.183 to 0.148, respectively.

Compared to MODIS (Moderate-resolution Imaging Spectroradiometer) AOD observations, the model free run shows an underestimation of the AOD values whereas the CALIOP assimilation corrects this underestimation and shows a quantitative good improvement in terms of AOD maps over the MB. The correlation between MODIS and the model (assimilation)- during the dust outbreak is 0.47 (0.52), whereas the bias is -0.18 (-0.02) and the RMSE is 0.36 (0.30).

The comparison of in-situ aircraft and balloon measurements to both modelled and assimilated outputs shows that the CALIOP lidar assimilation highly improves the model aerosol field. The evaluation with the LOAC (Light Optical Particle Counter) measurements indicates that the aerosol vertical profiles are well simulated by the direct model but with a general underestimation





of the aerosol number concentration especially in the altitude range 2–5 km. The CALIOP assimilation improves these results by a factor of 2.5 to 5.

Analysis of the vertical distribution of the desert aerosol concentration shows that the aerosol dust transport event is well captured by the model but with an underestimated intensity. The assimilation of CALIOP observations allows the improvement

of the geographical representation of the event within the model as well as its intensity by a factor of two in the altitude range 1–5 km.

# 1  Introduction

Aerosols play an important role in the atmospheric system of our planet. They have a significant impact on the Earth's radiation budget by direct scattering/absorption of sunlight and by changing cloud properties (e.g., Tegen and Lacis, 1996). Therefore,

they play a major role in the climate system. They also have an influence on the photochemistry of our atmosphere via the change in the intensity of photolysis of tropospheric oxydants (e.g., Tie et al., 2005). It is then important to well simulate the three-dimensional (3D) distribution of different types of aerosols within the chemistry and transport models (CTMs). Nevertheless, modelling of different types of aerosols is challenging due to the complexity of their physical/chemical transformations and the uncertainties in the parametrizations of their sources/sinks. Model improvements can be achieved by i) improving

physical parametrizations of the aerosols (see e.g., Sič et al., 2015), and/or ii) by assimilating aerosol products such as Aerosol Optical Depth (AOD) (e.g., Sič et al., 2016) or lidar backscatter/extinction profiles. Here, we deal with the improvement of modelled 3D distribution of aerosols using data assimilation.

Global observations of tropospheric aerosols have been performed from several satellite instruments including radiometers and lidars since late 1970s (the reader may refer to King et al. (1999) for an historical overview). The aerosol measurements

from these different instruments have provided many opportunities to study tropospheric aerosols on global and regional scales. This includes for example: MISR (The Multi-angle Imaging SpectroRadiometer ; Diner et al., 1989), POLDER (POLarization and Directionality of the Earth's Reflectances ; Deschamps et al., 1994), MODIS (MODerate resolution Imaging SpectroRadiometer ; Justice et al., 1998), MERIS (Medium Resolution Imaging Spectrometer ; Rast et al., 1999), SEVIRI (Spinning Enhanced Visible and Infrared Imager ; Aminou, 2002), and CALIOP (Cloud-Aerosol LIdar with Orthogonal Polarization ;

Winker et al., 2003).

Most of the aerosols related to air quality and pollution are found in the lower troposphere or boundary layer (e.g., Akimoto, 2003). These tropospheric aerosols are very important components of the Earth-Atmosphere-Ocean system and affect climate, pollution, air quality and health (see e.g., Ebi and McGregor, 2008). Research on air quality and pollution is generally based on: (i) the use of observations from different space or in-situ platforms or (ii) the use of numerical models that are used first to

better understand the physico-chemical processes that cause air pollution and then to predict the spatial and temporal evolution of the different types of aerosols. Nevertheless, current methodologies for detecting aerosols are based on observations from passive sensors (spectroradiometers) or active sensors (lidars). The spectroradiometers are generally characterized by a good horizontal coverage but with a very limited vertical resolution. On the contrary, lidar measurements have a very good vertical



resolution, but with a very low spatial coverage. On the other hand, modelling has the advantage of providing a 3D detailed spatio-temporal representation of different types of aerosols. Nevertheless, models generally face problems related mainly to the initial conditions, spatial resolution and emissions. Data assimilation is often used to overcome these difficulties and thus to improve the representation of different types of aerosols within models.

Chemical data assimilation consists in combining in an optimal way observations provided by instruments and a priori knowledge about a physical system such as model output. The observations are acting as constraints for the models, and thus can be used to overcome model deficiencies (e.g., El Amraoui et al., 2014). Typically, observation-minus-forecast (OMF) statistics are used for monitoring biases between the observations and the models (e.g., El Amraoui et al., 2010). Data assimilation systems produce a self-consistent four-dimensional (time and space) description of the dynamical and chemical state of the

atmosphere taking into account both the available chemical observations and our theoretical understanding of the atmospheric system.

    The assimilation of different aerosol components has been conducted in the framework of many studies including AOD (e.g. Rasch et al., 2001; Zhang and Reid, 2006; Niu et al., 2008; Benedetti et al., 2009; Liu et al., 2011; Shi et al., 2011), particulate matters (PMs) (e.g. Tombette et al., 2009; Lee et al., 2013) and lidar profiles (e.g., Sekiyama et al., 2010; Zhang et al.,

2011; Wang et al., 2014). Most of these studies have shown that data assimilation of different aerosol related quantities has a positive impact on aerosol forecasting, especially during the first forecast hours. Moreover, the assimilation of lidar profiles has the advantage of constraining the vertical structure of the model in a much more realistic and direct way. Consequently, the assimilation of lidar information would serve to reduce the influence of diffusion and better constrain the vertical structure (Campbell et al., 2010).

In this study, we present the assimilation module of lidar measurements in the CTM of Météo-France, MOCAGE (*Modèle de Chimie Atmosphérique à Grande Echelle*). The assimilation system coupled to the MOCAGE CTM was initially developed and used for atmospheric gases, predominately ozone ($O_3$) (e.g. Semane et al., 2007; El Amraoui et al., 2008b; Rabier et al., 2010; Bencherif et al., 2011; Barré et al., 2013; Emili et al., 2014; Hache et al., 2014; Abida et al., 2016), carbon monoxide (CO) (e.g. El Amraoui et al., 2010; Claeyman et al., 2010, 2011a; El Amraoui et al., 2014) and water vapour ($H_2O$) (e.g. Payra

et al., 2016). The assimilation of aerosol components within the MOCAGE CTM is more recent compared to that of chemical species. The first work related to the aerosol assimilation in the MOCAGE model concerned the MODIS AOD assimilation. The assimilation module presented in this work is an extension of the AOD assimilation system already described by Sič et al. (2016).

    We consider the extinction coefficient measurements from the CALIOP lidar on-board the CALIPSO (Cloud-Aerosol Lidar

and Infrared Pathfinder Satellite Observations) satellite. We focus on the African dust event occurred in late June–early July 2012 over the Mediterranean Basin (MB) during the TRAQA (*TRAnsport à longue distance et Qualité de l'Air dans le bassin méditerranéen*) field campaign held between June 26 and July 11, 2012 (see Section 3.3 later for more information about the TRAQA campaign). During this dust outbreak event, several aerosol profiles within the dust plume have been measured by the PCASP (Passive Cavity Aerosol Spectrometer Probe ; see Section 3.3.1) instrument on-board the instrumented aircraft.

This study aims principally to:



1. Present the lidar assimilation module as well as the first results dealing with the assimilation of CALIOP observations in terms of extinction coefficient into the MOCAGE CTM.

2. Evaluate the impact of lidar assimilation on the 3D tropospheric aerosol distribution at regional scale during this large scale event. The lidar measurements from the CALIOP instrument are assimilated into the MOCAGE CTM using the variational 3D-FGAT (First Guess at Appropriate Time) method. The impact of the CALIOP extinction coefficient assimilation on the aerosol distribution has been evaluated using a set of independent data including AERONET (AErosol RObotic NETwork), MODIS, aircraft as well as balloon measurements.

The paper outline is as follows: In Section 2 we describe the CALIOP lidar measurements which are assimilated in terms of extinction coefficient as well as the model and the assimilation system used in this study. Section 3 presents the independent observations used for the evaluation of CALIOP assimilation: AOD observations from MODIS and AERONET as well as the in-situ measurements collected during the TRAQA field campaign. Results concerning the assimilation of CALIOP lidar measurements during the TRAQA field campaign are presented in Section 4. Summary and conclusions are presented in Section 5.

## 2 Data and Analysis

### 2.1 Assimilated observations: CALIPSO/CALIOP measurements

The CALIPSO satellite is a partnership between NASA (National Aeronautics and Space Administration) and the French Space Agency, CNES (*Centre National d'Etudes Spatiales*). It was launched on April 28, 2006 with the cloud profiling radar system on the CloudSat satellite. It flew in the international "A-Train" constellation for coincident Earth observations until September 13, 2018 when CALIPSO began lowering its orbit from 705 km to 688 km above the Earth to resume formation flying with CloudSat as part of the "C-Train" (see: https://atrain.nasa.gov/). The CALIPSO satellite comprises three instruments, the CALIOP lidar, the IIR (Imaging Infrared Radiometer), and the WFC (Wide Field Camera). For more information on the CALIPSO measurements, the reader could refer to the website: https://www-calipso.larc.nasa.gov/.

The CALIPSO satellite provides new insight into the role that clouds and atmospheric aerosols (airborne particles) play in regulating Earth's weather, climate, and air quality (e.g., Huang et al., 2015). CALIOP is a two-wavelength lidar that has the ability to differentiate between types of aerosols. It provides the processed backscatter signal, and the retrieved backscattering and extinction coefficients. CALIOP is an elastic backscatter lidar operating at 532 nm and 1064 nm (Winker et al., 2010). It has generally two ascending and two descending orbits per day with a frequency of measurements that varies from day to day. CALIOP provides continuous measurements both in space and time but with limited horizontal coverage since CALIOP has a very narrow swath. Nevertheless, it allows to have aerosol and cloud profiles with a vertical resolution of 30 to 60 m (Winker et al., 2012). We will thus see the ability of the CALIOP aerosol observations to constrain the MOCAGE model and to provide added value when assessed against independent observations.



## 2.2 The model and assimilation system

MOCAGE (e.g., Josse et al., 2004; Teyssèdre et al., 2007) is a global 3D-CTM which covers the planetary boundary layer, the free troposphere and the stratosphere. It provides a number of optional configurations with varying domain geometries and resolutions, as well as chemical and physical parametrization packages. It includes several chemical schemes for stratospheric and tropospheric studies (e.g., Bousserez et al., 2007; Bencherif et al., 2011; Lacressonnière et al., 2012; Barré et al., 2013; Emili et al., 2014) . In this study, MOCAGE is forced dynamically by wind and temperature fields from the ARPEGE (*Action de Recherche Petite Echelle Grande Echelle*) model analyses, the global operational weather prediction model of Météo-France (Courtier et al., 1991). The model can include nested domains over smaller regions. In this study, the model is run in a two-domain configuration with a global grid of $2° \times 2°$ and a smaller nested domain with a grid of $0.2° \times 0.2°$ (called MEDI02) including the MB and the Sahara. The assimilation is done only on the nested domain MEDI02. The model uses a semi-Lagrangian transport scheme and includes 47 hybrid $(\sigma, P)$ levels from the surface up to $5\,hPa$, where $\sigma = P/P_s$; $P$ and $P_s$ are the pressure and the surface pressure, respectively. MOCAGE has a vertical resolution of about $800\,m$ in the vicinity of the tropopause and in the lower stratosphere, whereas in the boundary layer MOCAGE has seven levels with a vertical resolution between 40 and $400\,m$. In the free troposphere, the vertical resolution of MOCAGE varies from 400 to $800\,m$. Modelled aerosol species for this study are desert dust, sea salt, black carbon and primary organic carbon (Martet et al., 2009; Sič et al., 2015) . That represented aerosol species are far from complete, secondary aerosol which can be the major part of the fine fraction, being lacking. This partly explains the generally observed negative biaises observed in their study. Assimilation corrects simply this bias, but possibly also RMSE and correlation. The represented aerosol species in this study do not consider the secondary aerosols which can be the major part of the fine fraction. The lack of secondary aerosols may partly explain the negative biases generally observed in this study. The particle size distribution for each aerosol type is divided into six bins. The diameter range of different primary aerosol bins considered within the MOCAGE model is presented in Table 1. In total, we have 24 aerosol bins. Each aerosol bin is considered as a passive tracer during the model integration (including emission, transport and removal processes from the atmosphere). However, there are no physical transformations or chemical reactions between different types of aerosols/bins with gases. More details and information about the different parametrizations used within the MOCAGE CTM as well as the primary aerosols can be found in Sič et al. (2015).

The assimilation system is MOCAGE-Valentina (e.g., Emili et al., 2014; El Amraoui et al., 2014) which is an extension of the MOCAGE-PALM system (e.g., Massart et al., 2009) initially developed in the framework of the ASSET (ASSimilation of Envisat daTa) European project (Lahoz et al., 2007). This assimilation system is developed jointly by Météo-France and CERFACS (*Centre Européen de Recherche et de Formation Avancée en Calcul Scientifique*).

The assimilation system used in this study is the same as described in Sič et al. (2016). It uses the 3D-FGAT method which is a compromise between the 3D-Var and 4D-Var techniques (Fisher and Andersson, 2001). It compares the observation and background fields at the correct time and assumes that the increment to be added to the background state is constant over the entire assimilation window. The choice of this assimilation technique limits the size of the assimilation window, since it has to be short enough compared to chemistry and transport timescales. This technique has already produced good-quality results





compared to independent data, especially for $O_3$ (e.g. Semane et al., 2007; El Amraoui et al., 2008a, b; Rabier et al., 2010; Bencherif et al., 2011), CO (e.g. El Amraoui et al., 2010; Claeyman et al., 2011b), $H_2O$ (e.g. Payra et al., 2016) and AOD (e.g. Sič et al., 2016). This variant has the advantage that the linearised operator of the model evolution and its adjoint are replaced by the identity. The cost function of the 3D-FGAT incremental form is:

$$J(\delta x) = \frac{1}{2}(\delta x)^T \mathbf{B}^{-1}(\delta x) + \frac{1}{2}\sum_{i=1}^{p}(d_i - \mathbf{H}_i(\delta x))^T \mathbf{R}_i^{-1}(d_i - \mathbf{H}_i(\delta x)) \tag{1}$$

$\mathbf{B}$ and $\mathbf{R}$ are the background and the observation error covariance matrices, respectively.

In order to minimize the cost function more efficiently and to improve the convergence, the increment $\delta x$ is transformed to :

$$v = \mathbf{B}^{-1/2}\delta x. \tag{2}$$

In this way the cost function becomes :

$$J(x) = \frac{1}{2}v^{\mathrm{T}}v + \frac{1}{2}\sum_{i=1}^{N}(d_i - \mathbf{H}_i\mathbf{B}^{1/2}v)^{\mathrm{T}}\mathbf{R}_i^{-1}(d_i - \mathbf{H}_i\mathbf{B}^{1/2}v) \tag{3}$$

and its gradient :

$$\nabla J(\delta x) = v + (\mathbf{B}^{1/2})^T \sum_{i=1}^{N}\mathbf{H}_i^T\mathbf{R}_i^{-1}(d_i - \mathbf{H}_i\mathbf{B}^{1/2}v). \tag{4}$$

In this formulation, there is no need for the explicit specification of the inverse matrix $\mathbf{B}^{-1}$. Other advantages of such an approach are presented by Courtier et al. (1994).

The minimization of the cost function with the preconditioned form gives, as a result, an increment of the analysis in the space of variable $v$. After the minimization, it is necessary to pass into the model space again, and the increment is calculated as :

$$\delta x = \mathbf{B}^{1/2}v. \tag{5}$$

More details on the assimilation algorithm are described by Pannekoucke and Massart (2008) and Massart et al. (2012).

The background error covariance matrix $\mathbf{B}$ a matrix can be represented as :

$$\mathbf{B} = \Sigma\mathbf{C}\Sigma^T, \tag{6}$$

where $\Sigma$ is the diagonal matrix of the square root of the variances, and $\mathbf{C}$ is the positive definite symmetric matrix of horizontal and vertical correlations. For more information about the parametrization of the $\mathbf{C}$ matrix within MOCAGE-Valentina, the reader is referred to El Amraoui et al. (2014). The horizontal correlation is modelled using the integration of a generalized two-dimensional diffusion-type equation proposed by Weaver and Courtier (2001).

Note that $\mathbf{B}$ and $\mathbf{R}$ are estimated following the approach presented in El Amraoui et al. (2014) and Sič et al. (2016) and based on the Chi-square ($\chi^2$) optimisation. The background error variances, which are located on the diagonal of $\mathbf{B}$ matrix are found





to be 30% of the background state. For the observation error covariance matrix **R**, we only consider diagonal values, and they are found to be 15% of the observation values. Errors of observations are considered to be non-correlated, which means that all non-diagonal members (covariances) in the **R** matrix are zero.

In this study, the length scales are modelled using a Gaussian function in terms of geographic degrees for the horizontal
lengths (Pannekoucke and Massart, 2008), and in terms of pressure or number of model levels for the vertical lengths (Massart et al., 2009). The longitude and latitude lengths are constant and fixed to $0.2°$ which corresponds to a length scale of about 20–22 km.

Note that for a better model–observations comparison and memory optimisation, the assimilation cycle (assimilation window) is generaly divided into *time slots* of 1 hour. During each slot, observations are read, the observation operator is run, its
output field is interpolated to locations and times of the observations and compared with the observations, and the innovation vector is calculated and stored. In this study, the length of the assimilation window is the same as the time slot, consequently the cost function is minimized every hour.

## 2.3   Choice of the control variable for the aerosol assimilation

For aerosols, the modelled prognostic variable and observations are usually not the same physical quantity. In MOCAGE, the
prognostic variable is the aerosol mass concentration of each bin, and the quantities that we want to assimilate are the aerosol optical depth and the lidar backscatter/extinction profiles. For assimilation, it is necessary to choose the control variable $x$ (eq.3, 4) in the way to be the best adapted to our system and its purpose. One of the first criteria is that the control variable should be the same for all types of observations to be assimilated. The observation operator should be as simple as possible and easy to linearise.

In the literature we can find different choices for the control variable for the assimilation of different aerosol parameters :

  – Liu et al. (2011) used directly the prognostic variable in their model as the control variable.

  – Benedetti et al. (2009) used the 3D total aerosol concentration as the control variable. All bins have to be summed up, in order to obtain a new hybrid aerosol species.

  – Zhang et al. (2008) and Schroedter-Homscheidt et al. (2010) assimilated AOD which is used directly as the control
variable.

The different approaches have different impacts on the assimilation process. The first approach, the most ill-posed, is rigorous and straightforward, where all unknowns are considered as the control variable. This makes the control variables considerably bigger than in other approaches which can have a performance effect, in terms of memory optimisation and calculation time. Also, the **B** matrix has to include all bins separately with their variances and covariances which are difficult to define. But, the
analysis is partitioned to all bins automatically by the system, which searches for the optimal solution for all unknowns in the minimization of the cost function.





The second approach makes the control variable smaller than in the first approach. The cost function searches a solution in terms of only one variable (the 3D total aerosol concentration). The observational operator has first to sum up all bins. In order to linearise it, its tangent linear and adjoint have to contain information on how to partition the increment of analysis to all modelled bins.

The third approach is optimized for the assimilation of the AOD, where the observational operator would only do interpolation. The increment of analysis is also in quantities of AOD and after the minimization it is necessary to convert it into the 3D concentrations partitioned into the bins. The observation operator for the types of observations other than AOD, like lidar profiles, would be more complicated, especially for observations which are not column-integrated. Thus, the control variable would have to be changed from total column AOD, to the profile of AOD partial-column in order to preserve the information

in the vertical axis and avoid 2D/3D transformations and unnecessary hypothesis. Thus, this approach is not optimal for the assimilation of the observations that are not column-integrated.

Considering our needs to assimilated both AOD and lidar profiles within our system, we chose to use the 3D total aerosol concentration as the control variable as in Benedetti et al. (2009). With this choice, the problem of minimization of the cost function is better determined than in the first approach, where one observation would be used to constrain 30 unknowns (bins).

Also, it is better in terms of memory usage and computing performances. Still, in order to linearise the observation operator, it is necessary to make an assumption on how the analysis increment $\delta x^a$ will influence each bin.

In MOCAGE-Valentina, we keep the relative contribution of each bin constant in terms of their mass during the assimilation cycle. Bulk aerosol observations do not have any information of the contribution of different aerosol types. The validation of this approach has been done in Sič (2014), and successfully applied to AOD assimilation (Sič et al., 2016).

**2.4   Lidar assimilation**

The information on the aerosol vertical profile can be obtained from lidar observations. Incorporating this information in MOCAGE-Valentina is an important improvement in the model. For the assimilation of lidar profiles, it is necessary to develop an observation operator which links the total concentration in the model space with observed lidar quantities in the observation space. By using 3D total concentration as the control variable, we develop the system which is able to efficiently assimilate

AOD and lidar profiles. The theoretical concepts of the observation operator as well as the tangent-linear and adjoint tests concerning the lidar assimilation are presented in detail in "http://thesesups.ups-tlse.fr/2667/1/2014TOU30293.pdf"
We extend our study to the lidar measurements derived from the CALIOP instrument on-board the CALIPSO satellite and we focus on the TRAQA campaign for which we have access to a wide range of data-sets comprising AOD, in-situ measurements from the aircraft and the LOAC balloon observations. We study the case of a desert dust transport from Africa to the MB. The

added value of the assimilation of CALIOP measurements will be assessed in terms of the improvement of the representation of the desert aerosol within the MOCAGE model during this event.



## 3  Independent observations used for the evaluation of assimilated fields

### 3.1  MODIS

The MODIS instruments on-board the two EOS (Earth Observing System) satellites Terra (since 2000) and Aqua (since 2002), observe atmospheric aerosols and provide information about aerosol distribution on global coverage at horizontal resolu-
tions of 10 and 3 km. The evaluation of CALIPSO analyses in terms of MODIS AOD is done by using Collection C61 retrievals at 550 nm from both Terra and Aqua. The MODIS data concern both the deep blue and the dark target products. For more information about the improvements of the C61 collection in comparison to the C6 collection, the reader is referred to "https://modis-atmosphere.gsfc.nasa.gov/sites/default/files/ModAtmo/C061_Aerosol_Dark_Target_v2.pdf" for the deep-blue product, and to Gupta et al. (2016) for the dark target product.
The MODIS C61 version used in this comparison has a resolution of 10 km × 10 km. To fit the model resolution of $0.2° \times 0.2°$ over the MB in which the CALIOP assimilation has been performed, we calculate the so-called super-observations (Daley, 1993) obtained by averaging all MODIS observations within the model grid.

### 3.2  AERONET

The AERONET project is a federation of ground-based remote sensing aerosol networks. It uses CIMEL sun/sky radiometers
that make measurements within the 340-1020 nm for the direct sun radiation (Holben et al., 1998). For more than 25 years, the project has provided long-term, continuous and readily accessible public domain database of aerosol optical, microphysical and radiative properties for aerosol research and characterization, validation of satellite retrievals, and synergism with other databases. AERONET measurements are available at three levels: Level 1 (unscreened), Level 1.5 (cloud screened), and Level 2 (cloud screened and quality assured). The network imposes standardization of instruments, calibration, processing and distri-
bution. For more information about the AERONET project, the reader could be referred to "https://aeronet.gsfc.nasa.gov/". In this study, we used AERONET Level 2 (L2) data for the evaluation of model free run and AOD assimilated product.

### 3.3  In-situ measurements during the TRAQA field campaign

TRAQA is a scientific experiment within the MISTRALS (Mediterranean Integrated STudies at Regional And Local Scales) programme (http://www.mistrals-home.org). It was part of the preparation of the observation campaigns for the ChArMEx
component (Chemistry AeRosol Mediterranean EXperiment ; http://charmex.lsce.ipsl.fr). The ChArMEx project aimed at better estimating the impact of the chemical and particulate composition of the atmosphere on air quality and climate change at the scale of the MB (see, e.g. Jaidan et al., 2018).
The objectives of the TRAQA field campaign were to study transport, ageing and mixing of the pollution occurring in the MB (see, e.g. Basart et al., 2016). The aircraft flight domain was located over the North-Western MB during the summer 2012.
During the TRAQA campaign, between June 26 and July 12, 2012, several measurements of trace gases and aerosols were undertaken using numerous instruments such as the ATR-42 aircraft, atmospheric balloons (sounding and drift) and ground-





based instruments. Seven intensive observation periods (IOP) were performed using the ATR-42 aircraft operated by Météo-France/Safire. In particular, on June 29, 2012, a remarkable desert dust outbreak event marked by the recording of high values of AOD and aerosol concentrations was well captured by the aircraft instrument with a clear transport of aerosols to the MB (see, e.g. Sič et al., 2016).

In this study, we will focus on this desert dust outbreak to evaluate the added value of the CALIOP observations within the assimilation system compared to the free model run.

### 3.3.1  Aircraft measurement: PCASP

During TRAQA, the ATR-42 aircraft was equipped with the PCASP instrument. It is an aerosol spectrometer that measures the concentration and the particle size distribution of aerosols at high-frequency in 30 channels distributed over the diameter
range 0.1–3 μm (Strapp et al., 1992). Additional information on the instrument, the calibration methods and the measurement errors are reported by Cai et al. (2013). In this study we use data averaged over a one-minute interval with a spatial resolution of about 8 km.

### 3.3.2  LOAC

LOAC (Light Optical Particle Counter ; Renard et al., 2016) is an optical counter that measures the concentration in number of aerosols. It uses the two-angle diffusion aerosol measurement technique (Lurton et al., 2014; Renard et al., 2016). The
LOAC used during the TRAQA campaign has 20 size classes in the diameter range between 2 and 100 μm and installed on-board meteorological balloons. The number uncertainties for LOACs are in the order of 20 and 60 % for concentrations above 1 cm$^{-3}$ and for concentrations below 0.01 cm$^{-3}$, respectively. The vertical resolution of the LOAC measurements is the product of the LOAC time resolution including averaging, and the ballon ascent speed. It is ranging in the troposphere between
300 and 400 m, which is in the same range as the resolution of the model MOCAGE in the free troposphere.

## 4  Assimilation of CALIOP Lidar measurements during the TRAQA field campaign

We assimilate the extinction coefficient measurements from the CALIOP instrument into MOCAGE during the TRAQA campaign period. The objective is to assess the added value of CALIOP analyses compared to the model free run. Both fields are compared to different datasets presented in Section 3 including AOD observations from MODIS and AERONET as well as the
in-situ measurements collected during the TRAQA field campaign. The extinction coefficient observations from the CALIOP lidar are assimilated in the period between June 20 and July 11, 2012.

For this assimilation experiment, the domain in which the assimilation takes place, called the control domain, is defined with a resolution of $0.2° \times 0.2°$. It spatially covers the MB and the African Saharan desert. The boundaries of this control domain are : $[20°W - 40°E, 16°N - 52°N]$. The boundary conditions for both the model and the assimilation outputs are provided
by the global domain which is run with a spatial resolution of $2° \times 2°$.



## 4.1 Performance of the assimilation

To evaluate the impact of CALIOP lidar measurements on the modelled field we analyse the behaviour of the assimilation diagnostics in terms of observation minus analysis (OMA) and observation minus forecasts (OMF). Figure 1 shows the OMF and OMA histograms for all CALIOP lidar measurements in terms of extinction coefficient during the whole assimilation period (June 20–July 11, 2012). From this Figure, we notice that the OMA histogram is narrower and with its mean closer to zero than that for OMF (it means that the bias is reduced). The mean value of OMF (OMA) is 0.012 km$^{-1}$ (0.0095 km$^{-1}$) with a respective standard deviation of 0.15 km$^{-1}$ (0.14 km$^{-1}$). This indicates that the CALIOP lidar assimilated field is closer to the observations than the forecasts in terms of extinction coefficient. Note also that the bias between the observations and the model field is reduced after the assimilation process. The results from this a posteriori diagnostics show that the CALIOP assimilation has improved the model field since assimilated field is globally closer to the observations than the free model field.

## 4.2 Comparison with MODIS observations

In this Section, we present a first evaluation of the extinction coefficient assimilated product with respect to the MOCAGE free run by comparing both fields with the MODIS independent observations in terms of AOD.

Figure 2 presents a comparison between the AOD from the MOCAGE model and those from the CALIOP assimilation compared to MODIS over the MB for specific days of the whole desert dust outbreak event (from June 26 to July 1, 2012). All the fields presented in this figure were averaged over the day of comparison: the MOCAGE free run and the assimilated product are averaged within an hourly time resolution on a horizontal grid corresponding to that of the assimilation domain MEDI02. As for the free run and the assimilated fields, MODIS observations from AQUA and TERRA corresponding to the whole day of comparison are averaged on the grid of the model. In this figure, we also show the tracks of all CALIOP orbits performed during each day of comparison. It shows that the MB region is sounded every day by 2 to 3 descendant and ascendant orbits. Table 2 shows the statistics of such a comparison for all the days of the study. It shows the correlation, the bias and the RMSE between MODIS and the model free run on one hand and between MODIS and the assimilated product on the other hand. For all the comparison days, the statistics of the assimilated product are significantly improved compared to the free run field. For example, during the dust outbreak event of June 29, 2012, the correlation si improved from 0.47 to 0.52, whereas the bias (RMSE) is reduced from -0.18 (0.36) to -0.02 (0.30) for MODIS versus the direct free run and MODIS versus the assimilated field, respectively.

In figure 2, we note that during June 26, 2012 the event, is located between Northwest Africa covering Morocco to almost all of Spain and Portugal with a maximum of MODIS AOD values ranging between 0.5 and 1.2 over on the Atlantic Ocean. During June 27, the desert dust transport event moved north with a complete coverage of Spain and part of France with AOD values ranging from 0.4 to 0.8. During June 28, the event moved to the East toward Corsica with AOD values of about 0.5 recorded in the western part of the MB and part of Central Europe. During June 29 and June 30, the desert dust airmass covers a large part of the western Mediterranean with high AOD values exceeding 0.7 over Corsica and the Italian coasts. This period corresponds to the in-situ measurements performed during the TRAQA campaign (see Section 4.4 for more details).



Starting from July 1, 2012, we generally note that the event weakens progressively despite the fact that high AOD values still persist over Corsica. Note also that for all the days of comparison, AOD values from the free model run show a systematic underestimation of the desert dust amplitude. The assimilation of the lidar observations from the CALIOP instrument clearly improves the model field.

Many previous studies have highlighted the existence of biases between the CALIOP and MODIS observations (e.g., Kittaka et al., 2011; Redemann et al., 2012; Shikwambana and Sivakumar, 2018). The comparison between both datasets shows that MODIS AOD is generally higher than CALIOP-derived AOD (Oo and Holz, 2011). Ma et al. (2013) reported that the largest differences between CALIPSO and MODIS occurs during the active dust seasons over the major dust regions. Nevertheless, in our study the agreement between the AOD assimilated outputs and those resulting from the independent MODIS observations

is relatively good both in terms of quality and quantity. This shows that the assimilation of lidar observations has reduced the bias between MODIS and CALIOP data.

### 4.3 Comparison to AERONET observations

In this Section, we exploit the AOD in-situ observations from AERONET to quantify the added value of the CALIOP assimilated field in comparison to the model free run. We therefore use all available AERONET AOD L2 data collected during

the period of study from different stations located within the assimilation domain. Figure 3 shows the location as well as the number of the AOD observations for each measurement station used for the comparison during the considered period (from June 20 to July 11, 2012). In total, we consider measurements from 47 AERONET stations which are located in the domain of study. Most of the stations have a number of measurements greater than 300 (only 4 of the 47 used stations have a number of measurements less than 300 over the whole period of comparison). Moreover, the stations have been chosen to be represen-

tative of the whole domain. Modelled and assimilated fields available in each hour are interpolated into the AERONET time measurements. Moreover, in order to make the AOD wavelengths of different stations consistent with those of the model, we interpolate the AERONET data in a logarithmic scale at 550 nm by using all available neighbouring wavelengths (440, 500, 675 and 870 nm).

Figure 4 shows AOD time-series for selected stations illustrating the time evolution of the model free run and the CALIOP

assimilated fields in terms of AOD compared to the AERONET measurements between June 20 and July 11, 2012.

The stations located in the western part of the MB and Spain are marked by a dust episode of relatively high amplitude illustrated by high AOD values (arround June 27). The stations in Spain recorded the event earlier than the stations in France where it happened a few days later. This event is clearly highlighted by high AOD values. The localization as well as the duration of this event are well represented by both the model free run and the CALIOP analyses over all the stations of compar-

ison. Nevertheless, the AOD values from the free model run are underestimated over all the stations compared to AERONET measurements. The assimilated field corrects this underestimation regarding the AOD amplitude since the agreement between CALIOP analyses and AERONET data is better than that of the free run model. Table 3 presents correlation coefficient, bias and root mean square error (RMSE) between AERONET data and the model free run in one hand, and between AERONET and the assimilated field in the other hand over all the stations presented in Figure 3. Generally, the AOD derived from CALIOP



analyses present better statistics than the model free run compared to the AERONET data over all the stations. The comparison between AERONET data and the model output and between the AERONET data and the assimilated product is presented in Figure 5. This figure confirms again the overall conclusion about the improvement of the assimilation compared to the model free run. The improvement of the CALIOP assimilated field compared to the model free run is evidenced by the correlation

that increases whereas the bias and the RMSE are reduced. The correlation between AERONET observations and the model output (CALIOP assimilation) is 0.682 (0.753), whereas the bias is -0.063 (0.043) and the RMSE is 0.183 (0.148).

### 4.4   Comparison with the aircraft in-situ measurements

In this Section, we evaluate in detail the performance of CALIOP lidar assimilated field by comparing the results of assimilation and the MOCAGE model with the aerosol concentrations from in-situ measurements on-board the instrumented aircraft.

We therefore use measurements of the PCASP instrument that was embarked on-board the ATR-42 aircraft to measure the total concentration for particle diameters above 100 nm. Figure 6 shows the results of the total aerosol number concentration corresponding to the most representative flights which highlight the desert dust outbreak event over the MB already presented in Figure 2 : Flight A on June 29, 2012 from Toulouse to Corsica (Fig. 6-a), and flight B on the same day from Corsica to Toulouse (Fig. 6-b). Figure 6-1 shows the time evolution of the aerosol number concentration over the flight period. Figure 6-2

presents the aircraft altitude over the time flight from the departure to the arrival airports. In Figure 6-3, we present the map of the total AOD averaged over the flight period superimposed by the aircraft track with the departure (D) and the arrival (A) points for each flight. Figure 6-4 is the same as Figure 6-3 but for the mean value of the desert dust AOD over the flight period instead of the total AOD (obtained from all types of aerosols within the MOCAGE model).

    Flight A was performed on June 29 from 05 UTC to 09 UTC from Toulouse to Corsica. This flight coincides with the

beginning of the establishment of the desert dust event over southern France with incursions into eastern and north-eastern Spain and the western part of Italy (see Figure 6a-3-4). During this flight, three peaks of aerosol number concentrations were well captured by the aircraft with fairly high values throughout the flight from Toulouse to Corsica (maximum values varying between 8 cm$^{-3}$ and 14 cm$^{-3}$). These peaks were measured at altitudes between 4000 m and 5000 m. The contribution of the desert AOD to the total AOD exceeds 60% (Fig. 6-a-3 and Fig. 6-a-4). The MOCAGE free run clearly underestimates the

maximum values of these three peaks. However, the CALIOP assimilated field better represents the aerosol concentration peaks compared to the MOCAGE free run model. The assimilated product improves the field of the model and perfectly reproduces the cycle of variations in aerosol concentrations along flight A.

    During flight B (on June 29 between 10:00 am and 03:00 pm), the desert dust event is well established with an airmass of desert dust spreading over the MB from the east and north-east coasts of Spain to the coasts of Corsica and Italy (Figure 6b-3

and 6b-4). The contribution of the desert dust AOD to the total AOD exceeds 80%. Aircraft measurements show high values in terms of number aerosol concentrations during the majority of the flight throughout the desert dust airmass especially when the measurements are above 3000 m. The model free run fields highly underestimate the aircraft measurements while the CALIOP assimilation significantly improves this underestimation. They well reproduce aircraft measurements in terms of quantities and temporal variability.





These examples illustrate again the ability of the CALIOP assimilation to improve the model and then to reproduce the aircraft in-situ measurements in terms of aerosol concentrations. The assimilation of lidar aerosols data from CALIOP into MOCAGE improves the results by enhancing the aerosol number concentration by about a factor 2 getting closer to the aircraft measurements. Nevertheless, the minimum values of concentrations, close to zero, are not well reproduced by the MOCAGE

model. The general underestimation of MOCAGE and the assimilation compared to the independent aircraft measurements during the TRAQA aircraft campaign is likely due to the horizontal resolution of MOCAGE CTM (resolution of 0.2°) and the little number of lidar data around the studied aircraft domain due to the revisit time of CALIPSO. However, this comparison evidenced the added value in MOCAGE CTM using CALIOP aerosol measurements (see Figure 2).

### 4.5 Comparison with LOAC in-situ measurements

During the TRAQA campaign, the LOAC flew on-board three balloons, all launched from Martigues (5.05°E, 43.40°N : near Marseille France). We focus on the two flights performed on June 29, 2012 within the desert dust plume. The total horizontal extent of the LOAC is quite small ($\sim$ 15 km). This horizontal distance is smaller than the grid size of our domain of study ($\sim$ 20 km). Therefore, we assume that the LOAC measurements represent the vertical profile of the aerosol above the launch point. The LOAC two flights are launched at two different hours of the same day, in the morning and at noon, but they flew

within the same plume of desert dust (see the AOD maps on Figure 6). Figure 7 represents the vertical profile of the aerosol number concentration as deduced from the MOCAGE free run and the CALIOP assimilation both compared to the in-situ LOAC measurements of the two flights performed on June 29, 2012. The model free run well simulates the shape of the vertical profile, but for both cases, it underestimates the aerosol number concentration by a factor of 2.5 to 5 in the altitude range of 2–5 km. The assimilation of CALIOP lidar data improves this underestimation and shows a general good agreement

compared to LOAC measurements. The CALIOP lidar data assimilation product is closer to the LOAC measurements than that of the model free run especially in the altitude range of 1.5–5 km. This altitude range corresponds to the altitude within which the desert dust plume is transported (see section 4.6). CALIOP assimilation better simulates the shape of the profile as well as the aerosol number concentrations than the model free run.

The comparison of LOAC profiles to those resulting from the assimilation of AOD and CALIOP lidar extinction coefficient

observations (Figure 9 of Sič et al. (2016) for AOD assimilation, and Figure 7 of this study for CALIOP assimilation) seems to show an underestimation of the field resulting from the assimilation of the extinction coefficient of the CALIOP lidar. An explanation may be due to the fact that both MODIS AOD and LOAC measurements generally show an overestimation of aerosol concentrations compared to independent observations. Indeed, the study conducted by Shikwambana and Sivakumar (2018) highlights the overestimation of MODIS AOD compared to several datasets (e.g., CALIPSO, MERRA-2 and MISR). On

the other hand, the validation of LOAC measurements conducted by Renard et al. (2016) shows that the retrieved concentrations of the largest particles could be overestimated by up to 50% for particles above about 2 $\mu$m. Consequently, almost the total concentration of desert aerosols is affected by this overestimation since the majority of desert dust bins are greater than 2 $\mu$m (see Table 1).


## 4.6 Vertical structure of aerosol concentration

In this section, we evaluate the impact of assimilating the observations from the CALIOP instrument on the desert aerosol vertical distribution. Figure 8 shows an illustration of the impact of the assimilation of CALIOP observations on the vertical distribution of desert aerosol during the desert dust outbreak over the MB during June 29, 2012. Figure 8-a shows a measure-

ment orbit from the CALIOP instrument during June 29, 2012 between 12:45 pm and 12:58 pm (black and red colors). Fig 8-b shows the total attenuated backscatter ($km^{-1}$ $sr^{-1}$) at the wavelength of 532 nm from the CALIOP instrument corresponding to the measurements presented in Figure 8-a. The white rectangle shows part of the orbit indicated in red color in Figure 8-a. This part of the orbit highlights an airmass of desert dust above the MB in the altitude range between 1 and 5 km. This layer of desert dust is illustrated by relatively high values of the attenuated total backscatter from CALIOP measurements. Figure 8-c

shows the latitude cross-section (latitude versus altitude) of the desert dust aerosol along the blue line of Figure 8a (longitude = 7.1°) from the MOCAGE model. We note from this figure that the MOCAGE model provides the distribution of the desert dust concentration between 3 and 6 km above sea level at latitude 30°N and in the altitude range 1-5 km between latitudes 37°N and 49°N. The desert dust aerosol concentrations from the MOCAGE model are relatively low and do not clearly show the transport of desert dust as illustrated by the CALIOP measurements in Figure 8-b where the desert dust airmass is distributed

over a large region from latitude 34°N to almost latitude 45°N following the CALIOP orbit. Figure 8-d is the same as Figure 8-c but for the product resulting from the assimilation of CALIOP measurements. High values of the concentration of the desert aerosol are highlighted in the latitude range 38°N–44°N between 1 and 5 km of altitude. This figure shows a distribution of the desert dust aerosol above the MB much more realistic than the one from the free model run (see the comparison of the model and the analyses with MODIS independent data in Figure 2). The results presented in Figure 8 illustrate once again the ability

of the assimilation of the CALIOP product to improve the vertical distribution of the desert aerosol.

## 5    Summary and conclusion

The aim of this paper is to present and describe the assimilation of lidar observations from the CALIOP instrument in the chemistry-transport model (CTM) of Météo-France, MOCAGE (*MOdèle de Chimie Atmosphérique à Grande Échelle*). We presented the first results of the assimilation of the extinction coefficient observations of the CALIOP (Cloud-Aerosol Lidar

with Orthogonal Polarization) lidar instrument on-board the CALIPSO (Cloud-Aerosol Lidar and Infrared Pathfinder Satellite Observations) satellite during the TRAQA (*TRAnsport à longue distance et Qualité de l'Air dans le bassin méditerranéen*) field campaign. The assimilation system used in this study is an extension of the assimilation system developed for Aerosol Optical Depth (AOD) already presented by Sič et al. (2016). The methodology of assimilating different aerosol components (AOD or lidar profile) within the MOCAGE model consists in choosing the total concentration of aerosols as the control variable.

This approach has the advantage of making the problem of minimizing the cost function better determined than with other commonly used approaches (See e.g., Benedetti et al., 2009). Moreover, this approach is more adapted for the assimilation of various aerosol products such as lidar and AOD observations either independently or in synergy.



In this study, we have evaluated the added value of the assimilation of the CALIOP extinction coefficient observations to better document a desert dust transport event compared to the model free run. The CALIOP assimilation product has been evaluated against different independent datasets : AOD from MODIS (Moderate-resolution Imaging Spectroradiometer) and AERONET (AErosol RObotic NETwork), aerosol concentration from both the PCASP (Passive Cavity Aerosol Spectrometer

Probe) instrument on-board the ATR-42 aircraft and the LOAC (Light Optical Particle Counter) on-board the balloon during the TRAQA field campaign. The results show that CALIOP analyses improve the model compared to the AOD AERONET independent observations. The correlation is increased, while the bias and the RMSE (Root Mean Square Error) are reduced. The correlation between AERONET and the model free run (CALIOP assimilation) is 0.682 (0.753), whereas the bias is -0.063 (0.043) and the RMSE is 0.183 (0.148). Compared to MODIS observations, the model free run shows an underestimation of

the AOD values whereas the CALIOP assimilation improves this underestimation and shows a quantitative good improvement in terms of AOD maps over the MB. Compared to the LOAC in-situ measurements, the results showed that the assimilated field is closer to the measurements than the free run model field particularly in the altitude range $\sim$1.5–$\sim$5 km corresponding to the altitude range within which the desert dust plume is transported. Note that the represented aerosol species in this study do not consider the secondary aerosols which can be the major part of the fine fraction. The lack of the secondary aerosols may

partly explain the negative biases generally observed in this study.

Space-borne aerosol lidar observations have revealed to be useful for better understanding the aerosol properties in the atmosphere (e.g., Yu et al., 2010). Particularly, the CALIOP instrument offers many opportunities to better estimate the vertical distribution of aerosols (e.g., Winker et al., 2010). In this study we show that the assimilation of CALIOP lidar observations within the MOCAGE CTM allows a significant improvement in the model. We therefore get a better three-dimensional (3D)

distribution of aerosols in comparison to different independent observations.

Despite the fact that satellite nadir-view active sensors such as CALIOP have limited spatial coverage compared to passive sensors, the global observations of aerosol vertical distribution from lidars have contributed for improving the quality of atmospheric aerosol observations (IPCC, 2013). In addition, the assimilation of the lidar aerosol products in the MOCAGE CTM has some advantages. Compared to the assimilation of AOD observations, the assimilation of lidar profiles is more straightforward

and allows the introduction of direct information about the vertical distribution of aerosols into the model. This could give more realistic vertical aerosol distributions. Indeed, during the lidar assimilation, the minimization is done in each level where the observation is available independently of the other levels. Even the correlation between adjacent levels is done via the **B** matrix, the lidar assimilation will bring modifications according to the intensity and the quality of observations in each level. This has the advantage of better representing the different aerosol layers within the model and therefore better describing their

transport process (e.g., desert dust, biomass burning, volcanic ash, ...). On the contrary, the assimilation of the AOD will tend to uniformly modify the vertical profile of the model. This can induce biases, especially during extreme events. The assimilation of AOD and lidar profiles have been validated using the same versions of the model and the assimilation system. The next step will consist of making a complete comparison and a discussion about the results of both MODIS AOD and CALIOP lidar assimilations. We will particularly focus on the advantages and the limitations of each approach during a desert dust outbreak

event.





We also plan to study the added value of measurements from passive and active probes during volcanic eruption events. This is a very important theme for Météo-France since it is one of the VAAC (*Volcanic Ash Advisory Center*) whose responsibility extends over a large part of Europe, Asia and Africa.

As a perspective of this work, we will consider simultaneously assimilating the observations from passive and active sensors by carrying out an initial de-biasing of both observation datasets. A much more ambitious solution will consist to assimilate satellite radiances directly in a global model using an integrated approach. Assimilation of satellite radiances i.e. in numerical weather prediction assimilation systems has proved to be an essential component for improving the forecast skills, particularly for global models (e.g., Derber and Wu, 1998; McNally et al., 2000). This technique may be able to surpass some retrieval algorithms, and should provide improved results compared to data assimilation of retrieval products (e.g., Dong et al., 2007).





**Table 1.** Variation range of different primary aerosol bins within the MOCAGE model.

|  | Bin 1 | Bin 2 | Bin 3 | Bin 4 | Bin 5 | Bin 6 |
|---|---|---|---|---|---|---|
| Desert dust ($\mu$m) | 0.1–1 | 1–2.5 | 2.5–5 | 5–10 | 10–30 | 30–100 |
| Sea salt ($\mu$m) | 0.003–0.13 | 0.13–0.3 | 0.3–1 | 1–2.5 | 2.5–10 | 10–20 |
| Black carbon ($\mu$m) | 0.0001–0.001 | 0.001–0.003 | 0.003–0.2 | 0.2–1 | 1–2.5 | 2.5–10 |
| Organic carbon ($\mu$m) | 0.0005–0.003 | 0.003–0.1 | 0.1–0.3 | 0.3–1 | 1–2.5 | 2.5–10 |





**Table 2.** Statistics (Correlation, bias and RMSE) between MODIS observations and MOCAGE free run/assimilation during the TRAQA campaign between June 20 and July 11, 2012.

|  | MOCAGE free run | | | MOCAGE assimilation | | |
|---|---|---|---|---|---|---|
|  | Correlation | Bias | RMSE | Correlation | Bias | RMSE |
| June 20 | 0.44 | -0.17 | 0.34 | 0.53 | -0.06 | 0.29 |
| June 21 | 0.56 | -0.14 | 0.27 | 0.58 | -0.01 | 0.25 |
| June 22 | 0.42 | -0.13 | 0.34 | 0.50 | -0.01 | 0.33 |
| June 23 | 0.39 | -0.17 | 0.38 | 0.47 | -0.03 | 0.33 |
| June 24 | 0.40 | -0.23 | 0.47 | 0.44 | -0.08 | 0.41 |
| June 25 | 0.32 | -0.25 | 0.54 | 0.37 | -0.10 | 0.48 |
| June 26 | 0.27 | -0.22 | 0.51 | 0.39 | -0.04 | 0.45 |
| June 27 | 0.32 | -0.24 | 0.49 | 0.41 | -0.06 | 0.42 |
| June 28 | 0.29 | -0.21 | 0.41 | 0.40 | -0.04 | 0.34 |
| June 29 | 0.47 | -0.18 | 0.36 | 0.52 | -0.02 | 0.30 |
| June 30 | 0.28 | -0.18 | 0.37 | 0.41 | -0.01 | 0.31 |
| July 1 | 0.45 | -0.15 | 0.30 | 0.53 | 0.02 | 0.26 |
| July 2 | 0.44 | -0.15 | 0.28 | 0.51 | 0.02 | 0.25 |
| July 3 | 0.50 | -0.15 | 0.27 | 0.56 | 0.04 | 0.26 |
| July 4 | 0.42 | -0.13 | 0.28 | 0.49 | 0.03 | 0.29 |
| July 5 | 0.45 | -0.14 | 0.28 | 0.51 | 0.02 | 0.27 |
| July 6 | 0.49 | -0.17 | 0.28 | 0.56 | 0.01 | 0.24 |
| July 7 | 0.30 | -0.19 | 0.29 | 0.46 | -0.05 | 0.22 |
| July 8 | 0.46 | -0.18 | 0.27 | 0.51 | -0.02 | 0.20 |
| July 9 | 0.43 | -0.15 | 0.22 | 0.52 | 0.03 | 0.19 |
| July 10 | 0.48 | -0.13 | 0.23 | 0.56 | 0.06 | 0.21 |
| July 11 | 0.60 | -0.12 | 0.22 | 0.63 | 0.07 | 0.25 |
| All days | 0.37 | -0.17 | 0.35 | 0.45 | -0.01 | 0.31 |



**Table 3.** Correlation, bias and root mean square error (RMSE) between AERONET observations and MOCAGE free/assimilation runs

| Station ( Lat (°) ; Lon (°) ) | Altitude (m) | $N_{obs}$ | MOCAGE free run | | | CALIOP Assimilation | | |
|---|---|---|---|---|---|---|---|---|
| | | | Correlation | Bias | RMSE | Correlation | Bias | RMSE |
| Villefranche ( 43.684 ; 7.329 ) | 130.0 | 387 | 0.61 | 0.08 | 0.13 | 0.74 | -0.01 | 0.08 |
| Davos ( 46.813 ; 9.844 ) | 1596.0 | 270 | 0.55 | 0.06 | 0.09 | 0.51 | -0.02 | 0.09 |
| Granada ( 37.164 ; -3.605 ) | 680.0 | 1025 | 0.72 | 0.04 | 0.13 | 0.82 | -0.06 | 0.11 |
| Laegeren ( 47.48 ; 8.351 ) | 735.0 | 306 | 0.62 | 0.06 | 0.11 | 0.68 | -0.01 | 0.09 |
| Messina ( 38.197 ; 15.567 ) | 15.0 | 662 | 0.51 | 0.06 | 0.11 | 0.78 | -0.08 | 0.11 |
| Evora ( 38.568 ; -7.912 ) | 293.0 | 1029 | 0.83 | 0.0 | 0.12 | 0.9 | -0.07 | 0.1 |
| Lampedusa ( 35.517 ; 12.632 ) | 45.0 | 1242 | 0.61 | 0.05 | 0.12 | 0.85 | -0.08 | 0.11 |
| Tamanrasset INM ( 22.79 ; 5.53 ) | 1377.0 | 945 | 0.45 | 0.09 | 0.32 | 0.5 | -0.05 | 0.3 |
| Oujda ( 34.653 ; -1.899 ) | 620.0 | 580 | 0.68 | 0.16 | 0.19 | 0.7 | -0.04 | 0.2 |
| Tabernas PSA-DLR ( 37.091 ; -2.358 ) | 500.0 | 1029 | 0.78 | 0.12 | 0.17 | 0.83 | -0.01 | 0.1 |
| Porquerolles ( 43.001 ; 6.161 ) | 22.0 | 625 | 0.78 | -0.0 | 0.08 | 0.88 | -0.09 | 0.1 |
| Lecce University ( 40.335 ; 18.111 ) | 30.0 | 849 | 0.32 | 0.07 | 0.1 | 0.51 | -0.06 | 0.1 |
| Cabo da Roca ( 38.783 ; -9.5 ) | 140.0 | 218 | 0.95 | 0.02 | 0.16 | 0.97 | -0.05 | 0.11 |
| Forth Crete ( 35.333 ; 25.282 ) | 20.0 | 297 | -0.27 | 0.05 | 0.09 | 0.42 | -0.03 | 0.07 |
| Sede Boker ( 30.855 ; 34.782 ) | 480.0 | 1047 | 0.17 | 0.01 | 0.08 | 0.47 | -0.11 | 0.13 |
| Limassol Cut-Tepak ( 34.675 ; 33.043 ) | 22.0 | 891 | 0.28 | 0.09 | 0.12 | 0.29 | -0.04 | 0.1 |
| Palencia ( 41.989 ; -4.516 ) | 750.0 | 825 | 0.85 | 0.02 | 0.09 | 0.89 | -0.05 | 0.07 |
| Huelva ( 37.016 ; -6.569 ) | 25.0 | 1131 | 0.8 | 0.02 | 0.15 | 0.86 | -0.08 | 0.13 |
| Izana ( 28.309 ; -16.499 ) | 2391.0 | 1184 | 0.74 | -0.03 | 0.16 | 0.84 | -0.17 | 0.21 |
| Ersa ( 43.004 ; 9.359 ) | 80.0 | 757 | 0.77 | 0.04 | 0.11 | 0.9 | -0.06 | 0.09 |
| Sagres ( 37.048 ; -8.874 ) | 26.0 | 546 | 0.9 | 0.02 | 0.18 | 0.91 | -0.07 | 0.14 |
| Cerro Poyos ( 37.108 ; -3.487 ) | 1830.0 | 116 | 0.73 | -0.05 | 0.07 | 0.74 | -0.17 | 0.18 |
| Santa Cruz Tenerife ( 28.473 ; -16.247 ) | 52.0 | 1045 | 0.62 | 0.19 | 0.51 | 0.75 | 0.01 | 0.41 |
| Malaga ( 36.715 ; -4.478 ) | 40.0 | 969 | 0.75 | 0.1 | 0.17 | 0.85 | -0.02 | 0.09 |
| Nes Ziona ( 31.922 ; 34.789 ) | 40.0 | 617 | 0.24 | 0.05 | 0.11 | 0.52 | -0.08 | 0.12 |
| Venise ( 45.314 ; 12.508 ) | 10.0 | 1166 | 0.72 | 0.11 | 0.14 | 0.72 | -0.02 | 0.09 |
| OHP Observatoire ( 43.935 ; 5.71 ) | 680.0 | 759 | 0.74 | 0.01 | 0.09 | 0.84 | -0.06 | 0.08 |
| Carpentras ( 44.083 ; 5.058 ) | 100.0 | 796 | 0.75 | 0.02 | 0.09 | 0.85 | -0.05 | 0.08 |
| San Giuliano ( 42.286 ; 9.519 ) | 10.0 | 700 | 0.63 | 0.09 | 0.15 | 0.83 | -0.02 | 0.08 |
| Cairo EMA ( 30.081 ; 31.29 ) | 70.0 | 461 | 0.06 | 0.15 | 0.19 | 0.11 | 0.04 | 0.12 |
| Athens-NOA ( 37.988 ; 23.775 ) | 130.0 | 105 | 0.51 | 0.19 | 0.19 | 0.58 | 0.05 | 0.08 |
| Calern OCA ( 43.749 ; 6.927 ) | 1270.0 | 692 | 0.78 | 0.0 | 0.09 | 0.88 | -0.08 | 0.09 |
| Tizi Ouzou ( 36.699 ; 4.056 ) | 133.0 | 315 | 0.91 | 0.16 | 0.19 | 0.9 | 0.02 | 0.08 |
| Thessaloniki ( 40.63 ; 22.96 ) | 60.0 | 1094 | 0.54 | 0.12 | 0.15 | 0.61 | 0.0 | 0.09 |
| Madrid ( 40.452 ; -3.724 ) | 680.0 | 990 | 0.75 | 0.01 | 0.09 | 0.84 | -0.06 | 0.09 |
| Aubiere LAMP ( 45.761 ; 3.111 ) | 423.0 | 307 | 0.56 | 0.03 | 0.1 | 0.65 | -0.05 | 0.1 |
| Zaragoza ( 41.633 ; -0.882 ) | 250.0 | 984 | 0.72 | 0.05 | 0.1 | 0.78 | -0.03 | 0.07 |
| Barcelona ( 41.386 ; 2.117 ) | 125.0 | 453 | 0.8 | 0.1 | 0.16 | 0.88 | -0.01 | 0.09 |
| Frioul ( 43.266 ; 5.293 ) | 40.0 | 709 | 0.87 | 0.03 | 0.1 | 0.92 | -0.06 | 0.08 |
| Palma de Mallorca ( 39.553 ; 2.625 ) | 10.0 | 1006 | 0.76 | 0.11 | 0.15 | 0.82 | -0.02 | 0.09 |
| Montsec ( 42.051 ; 0.73 ) | 1574.0 | 621 | 0.68 | 0.01 | 0.07 | 0.78 | -0.08 | 0.1 |
| La Laguna ( 28.482 ; -16.321 ) | 568.0 | 666 | 0.72 | 0.34 | 0.61 | 0.81 | 0.13 | 0.44 |
| Autilla ( 41.997 ; -4.603 ) | 873.0 | 738 | 0.78 | 0.0 | 0.08 | 0.9 | -0.06 | 0.08 |
| Avignon ( 43.933 ; 4.878 ) | 32.0 | 878 | 0.84 | 0.02 | 0.09 | 0.9 | -0.06 | 0.08 |
| Ouarzazate ( 30.928 ; -6.913 ) | 1136.0 | 548 | 0.46 | 0.21 | 0.31 | 0.76 | 0.03 | 0.17 |
| Burjassot ( 39.508 ; -0.418 ) | 30.0 | 738 | 0.69 | 0.09 | 0.17 | 0.76 | -0.02 | 0.11 |
| Rome Tor Vergata ( 41.84 ; 12.647 ) | 130.0 | 912 | 0.59 | 0.04 | 0.08 | 0.75 | -0.08 | 0.11 |
| All sites | | 34230 | 0.677 | -0.066 | 0.185 | 0.746 | 0.046 | 0.149 |



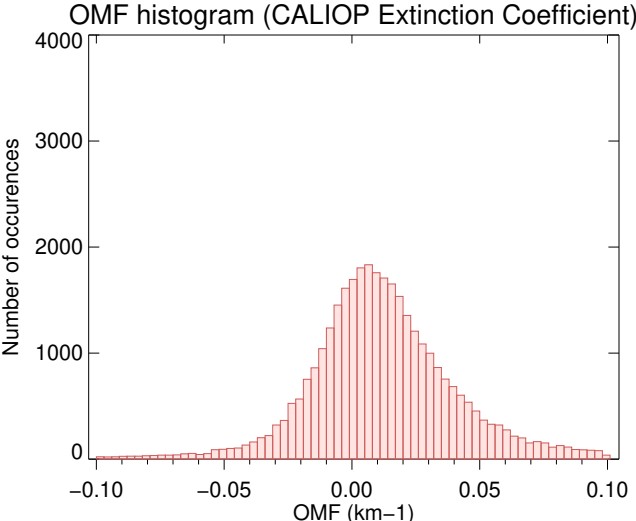

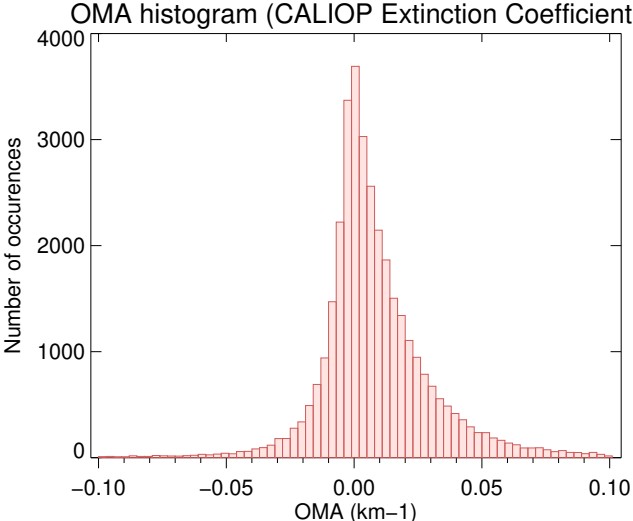

**Figure 1.** Histograms of the assimilation diagnostics in terms of OMF (observation minus forecast) in top and OMA (observation minus analysis) in bottom.



**Figure 2.** Comparison of Aerosol Optical Depth obtained by both the MOCAGE free run model **(a)** and the CALIOP assimilation **(c)** to the MODIS product from both Aqua and Terra for specific days **(b)** between June 26 and July 1, 2012 (from up to bottom). Figures in **(d)** show the tracks of all CALIOP orbits performed during each day of comparison.



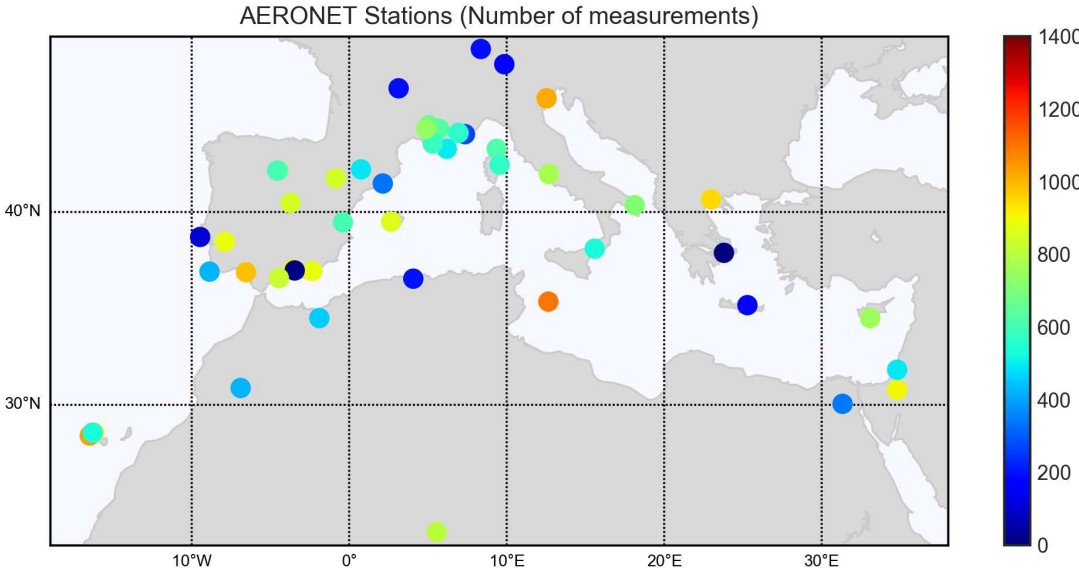

**Figure 3.** Map of the AERONET stations used for the validation of CALIOP assimilation. The color code presents the number of observations in each station used within the whole period of study.

**Figure 4.** Time series of AOD at 550 nm of the AERONET data (black), the free run model (blue) and the CALIOP assimilation (red) from June 20 to July 11, 2012 for specific AERONET stations. The name as well as the coordinates (longitude and latitude) of the specified station is marked at the top of each panel. Correlation, bias and root mean square error for both the direct model and the assimilation run as compared to the AERONET data are given in Table 3.



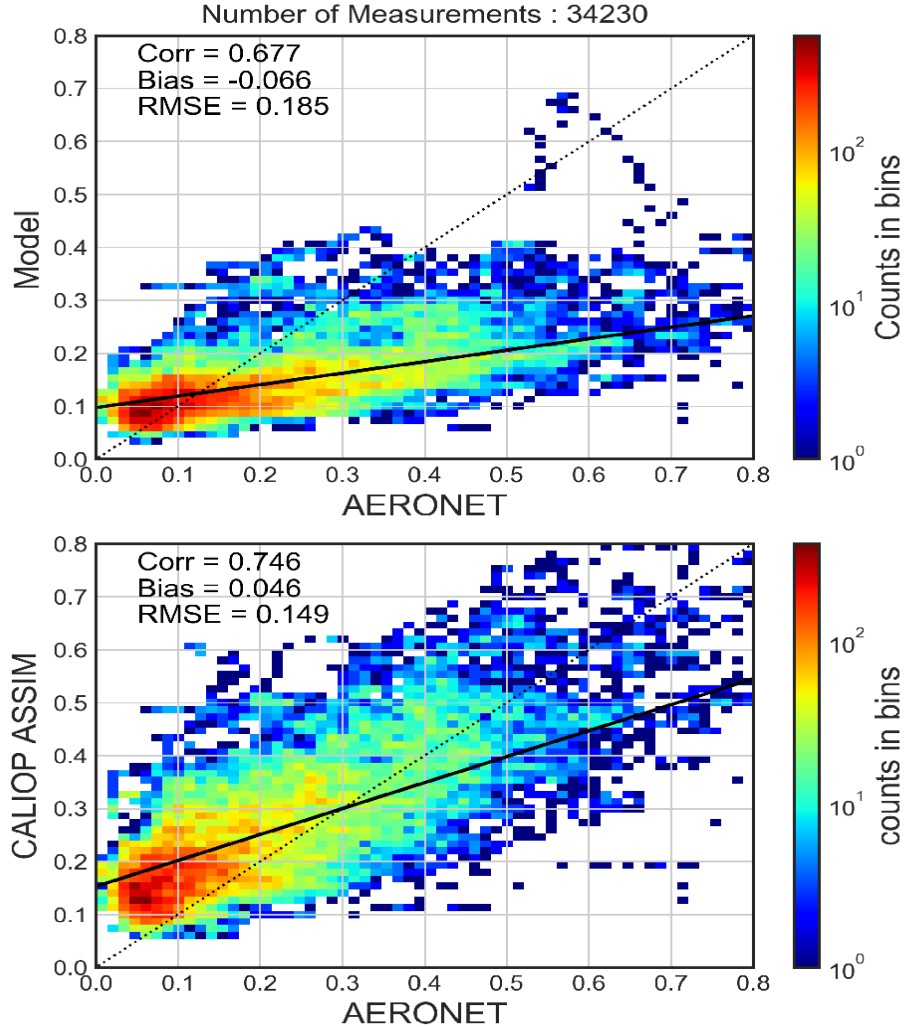

**Figure 5.** Scatter plots of AOD where colours represent the number of counts between the independent observations AERONET and the two simulations: The direct free model run (top), and the CALIOP assimilation (bottom). Correlation, bias and root mean square error are noted in each panel. The presented data correspond to the period of comparison from June 20 until July 11, 2012, and cover all the AERONET stations presented in Figure 4.



**Figure 6.** **(1)**: Time evolution of aerosol number concentration (in $cm^{-3}$) measured by the PCASP instrument on board the ATR aircraft (black) compared with the free model run (blue) and the CALIOP assimilation (red ). **(2)**: The aircraft altitude (in km) during its trajectory from the departure to the arrival points. **(3)**: The total AOD averaged over the time of flight deduced from the CALIOP assimilation field. The thick black line represents the aircraft trajectory from the departure (D) to the arrival (A). **(4)**: same as for **(3)** but for the desert dust aerosol. All these figures are presented for the two different flights: **(a)**: the flight on June 29, 2012 between 5 am and 9 am , and **(b)**: the flight on June 29, 2012 between 10 am and 15 pm.



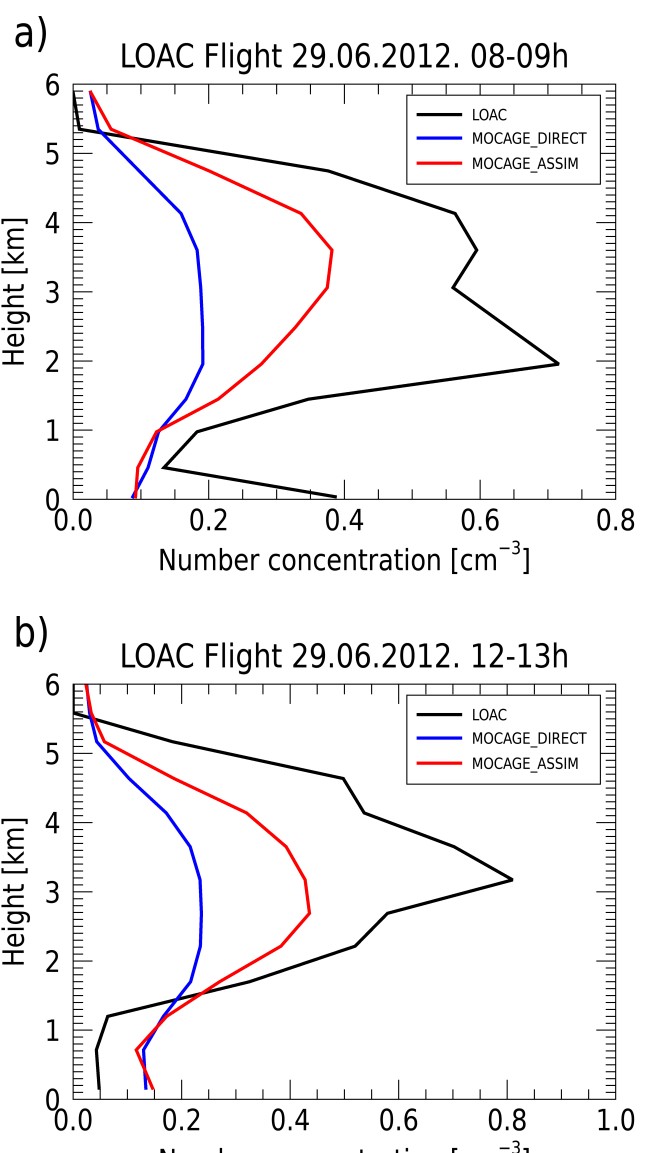

**Figure 7.** Vertical profile of aerosol number concentration in $cm^{-3}$ obtained by the model free run (blue) and the CALIOP assimilation (red) both compared to the LOAC measurements (black). The comparison is done for two LOAC flights both performed on June 29, 2012: between 8 am and 9 am (**a**), and between 12 pm and 13 pm (**b**).



**Figure 8.** **(a)** : A measurement orbit of the CALIOP instrument during June 29, 2012 between 12:45 pm and 12:58 pm (red color corresponds to the part of the orbit which coincides with the desert dust airmass over the MB). **(b)** : The vertical profile of the total attenuated backscatter ($km^{-1}$ $sr^{-1}$) as measured by CALIOP at the wavelength of 532 nm corresponding to the orbit presented in Figure 8-a. **(c)** : The aerosol classification corresponding to the vertical profiles showed in Figure 8-b (the majority of the aerosols are of dust origin). **(d)** : Vertical profiles of desert dust concentration in kg m$^{-3}$ issued from MOCAGE model and corresponding to the CALIOP orbit of Figure 8-a issued from MOCAGE and CALIOP assimilation. **(e)** : Same as Figure 8-d but for the assimilation of the extinction coefficient from CALIOP lidar instrument. The white rectangle in **(b)**, **(c)**, **(d)** and **(e)** corresponds to the part of the orbit indicated in red color in **(a)**.



*Acknowledgements.* This work is funded in France by the Centre National de Recherches Météorologiques (CNRM) of Météo-France and the Centre National de Recherches Scientifiques (CNRS). We would also like to thank NASA and the ICARE, the French atmospheric composition database (CNES and CNRS-INSU) for providing the AERONET data. ¨



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
