# Peer review of "Aerosol data assimilation in the chemical transport model MOCAGE during the TRAQA/ChArMEx campaign: Lidar observations"

_Atmospheric Measurement Techniques, 2019_

## Referee Comment (RC1) · Anonymous Referee #1 · 4 Mar 2020

General Comments: In this manuscript, the authors present the first assimilation of aerosol extinction coefficients measured by the CALIOP in the model called MOCAGE during the TRAQA/ChArMEx campaign. As expected, the assimilation of CALIOP aerosol vertical observations contributes to constrain the model simulated aerosol vertical distributions. General speaking, the manuscript is scientifically sound and well organized. I recommend accepting it after addressing the following comments. Major comments: 1) The detail information about the assimilated CALIOP observation data is missed. Also, the CALIOP retrieved aerosol extinction coefficients are generally contaminated by cloud. To eliminate the assimilation of the bad observations, the quality control of the CALIOP aerosol retrievals is generally required. See Cheng et al,

(https://doi.org/10.5194/acp-19-13445-2019) 2) The uncertainties of the background state (B) and CALIOP observation (R) will significantly affect the assimilation results, however, the B and R are artificially assumed as 30% and 15% respectively in the assimilation system without any explanation. 3) The control variable is the total aerosol mass concentration in your assimilation system. You should explain how to convert the control variable to the aerosol extinction coefficients in the manuscript. Specific comments: 1. P5L32 What do you mean about "the increment to be added to the background state is constant over the entire assimilation window"? As my understanding, the assimilation window in your experiment only has one time slot. 2. Please also explain the di in the formula 1. 3. P6L25 and P7L4 Do you mean the two-dimensional diffusion-type equation is the Gaussian function? 4. P7L6 What are the longitude and latitude lengths? Do you only assimilate only the observations within about 20–22 km? If so, it looks the horizontal lengths are too small. How about the vertical lengths? 5. P7L9 generaly → generally 6. P7L15 you did not assimilate the AOD in your experiment. 7. P7L17 What do you mean about "the control variable should be the same for all types of observations to be assimilated"? 8. Figure 1, it looks the assimilation system are more effective when the OMF is negative. This probably corresponds to the lower observations and lower observation uncertainties. So the assimilation system give more trust to the observation. 9. Figure 5, the simulated AODs of both the free run and assimilation experiment are overestimated when the AODs are lower than 0.1. This probably due to the observations from the sites located at high altitude such as >1km. 10. Figure 8, You compare the simulated aerosol concentrations in the free run and assimilation, however, it is difficult to judge which is better since you do not have the observations. I recommend you compare the simulated extinction coefficients with the CALIOP observations.

---

## Referee Comment (RC2) · Anonymous Referee #2 · 17 Mar 2020

I - General comments

This manuscript aims to present, for a case study of a desert dust outbreak in June 2012, the impact of the assimilation of LIDAR observation on a simulation performed with the chemistry transport model MOCAGE. This manuscript has a companion paper (amt-2016-60) focused on the assimilation of aerosol optical depth from MODIS.

Its a topic of scientific interest and within the scope of Atmospheric Measurement Techniques due to the large and rich set of aerosol observations from multiple instruments used to perform the analysis. The presentation of the work is globally very clearly organized, but appears sometimes too limited to a simple superficial description. A more

in depth presentation of the results and of some crucial choices would be very useful to improve the added value of this manuscript. In particular, I have been deeply surprised to find only in the conclusion a mention of a comparison between the previous analysis reported in the companion paper. From my point of view this should be included in the present manuscript.

On the other hand a large part of the text, providing the general description of the assimilation method (globally not new), the model (not new), the observation database used (not new) which is common with the companion paper could be certainly limited.

II - Detailed Comments

The sections 2.2 and 2.3 are largely common with the sections 2 and 3 of the companion paper. They could be largely reduced (or at least put in annex) to focus on the change performed to assimilate LIDAR observation. However, emission used should be mentioned.

p7, line 1 and 2 : The determination of the background error variances and the observation error covariance should be on contrary more detailed. These are results specific to the case study presented.

p7, line 4-7 : This paragraph should be more detailed also. What is the vertical correlation length? How the link between observation with a vertical resolution of 30 to 60 m and model with a vertical resolution of 400 to 800 m is made?

Section 3 : I suggest to merge each paragraph of this section with the corresponding one in the section 4 to avoid a dilution effect for the readers.

section 4.1 : Could the definition of observation minus analysis and observation minus forecast be recalled? I understand it is verification of the good behavior of the minimization process. In this case, it could be simply presented like this. Moreover, I think it would be interesting to compare the free forecast and the analysis also in terms of AOD. It could then be more directly compared to the results obtained with the

measurements data from MODIS and AERONET.

p12, line 11: "...bias between MOCAGE and MODIS data." I guess?

p13, line 5-6: It is clear that the assimilation of LIDAR data allows a better representation of high values. However, I believe that a comment here on the fact that the positive bias for low values is increased (as shown in figure 5) would be welcome.

figure 6 and figure 7: It appears the number concentrations values observed with the ballon flight ($\sim$0.5 cm-3) are much lower than those observed with PCSAP instrument on board the ATR aircraft in the desert dust plume ($\sim$5 cm-3). Any comments on this?

figure 7: The horizontal axes could be set to the same scale to ease the comparison between the two flight.

p8, line 12 : assimilated <-> assimilate

p8, line 26 : "http://thesesups.ups-tlse.fr/2667/1/2014TOU30293.pdf" <-> Sic (2014)

p11, line 24 si <-> is

p34, line 5 the reference to the companion paper should be updated.
* * *

---

## Author Comment (AC1) · 4 Jun 2020

The comment was uploaded in the form of a supplement:
https://www.atmos-meas-tech-discuss.net/amt-2019-482/amt-2019-482-AC1-supplement.pdf

---

## Author Response (AR1)

**Aerosol data assimilation in the chemical transport model MOCAGE during the TRAQA/ChArMEx campaign: Lidar observation**

El Amraoui, L., Sič, B., Piacentini, A., Marécal, V., Attié, J.-L., and Frebourg, N.

Paper under review for : Atmospheric Measurements and Techniques (amt-2019-482).

First of all, we are very grateful for the valuable comments of the reviewers. Bellow we answer point by point to their remarks.

Note that all the new changes with respect to the first submission are indicated in boldface in the new version of the paper.

**Response to the comments of the Anonymous Referee #1**

**General comments :**

In this manuscript, the authors present the first assimilation of aerosol extinction coefficients measured by the CALIOP in the model called MOCAGE during the TRAQA/ChArMEx campaign. As expected, the assimilation of CALIOP aerosol vertical observations contributes to constrain the model simulated aerosol vertical distributions. General speaking, the manuscript is scientifically sound and well organized. I recommend accepting it after addressing the following comments.

**Major comments :**

1) The detail information about the assimilated CALIOP observation data is missed. Also, the CALIOP retrieved aerosol extinction coefficients are generally contaminated by cloud. To eliminate the assimilation of the bad observations, the quality control of the CALIOP aerosol retrievals is generally required. See Cheng et al, ([https://doi.org/10.5194/acp-19-13445-2019](https://doi.org/10.5194/acp-19-13445-2019))

➔ **Information related to the selection of CALIOP data (quality flag and cloud contamination) were taken into account when we assimilated the extinction profiles from the CALIOP instrument. This has been clarified in the new version of the paper with the reference to the publication of _Cheng et al. 2019_.**

2) The uncertainties of the background state (B) and CALIOP observation (R) will significantly affect the assimilation results, however, the B and R are artificially assumed as 30% and 15% respectively in the assimilation system without any explanation.

➔ **In data assimilation, the covariance matrices B and R should be consistent (see i.e., _Talagrand, 2003_). This consistency could be ensured thanks to the help of the Chi2 test (_Lahoz et al., 2007_; _Ménard and Chang, 2000_). A chi2 value close to 1 is a good indication of the consistency of the assimilation algorithm (_Talagrand et al., 2003_). Therefore, if B is known, the chi2 test can give information about the R matrix (see, _El Amraoui et al., 2014_).**
**The B matrix estimation concerning the aerosol assimilation has been already presented by _Sič et al., 2016_. Based on this estimation of the B matrix, we have estimated the R matrix using the chi2 test in order to check the consistency of the assimilation algorithm, particularly B and R matrices in the same way as we already done in previous studies (e.g., _El Amraoui et al., 2014_).**
**These information have been clarified in the revised version of the paper.**

3) The control variable is the total aerosol mass concentration in your assimilation system. You should explain how to convert the control variable to the aerosol extinction coefficients in the manuscript.

➔ **For data assimilation of lidar profiles, the observation operator transforms the control variable into the lidar observed quantities. It makes the link between the total concentration of aerosols defined in the model space with the observed lidar quantities situated in the observation space.**
**First, the lidar profile observation operator within the MOCAGE-PALM assimilation system sums all individual aerosol species in order to calculate the total concentration.**

Second, it solves the lidar equation by taking into account the contributions of aerosols, gases and Rayleigh scattering. In order to make connection between total aerosol mass and lidar observed quantities, the relative mass contributions among aerosol species and sizes (bins) are considered constant in the tangent linear and adjoint operators (during an assimilation cycle).

At the end of the cycle, to calculate an increment, the same relative mass contribution determined before the assimilation is convert the total concentration into the all aerosols bins.

The observation operator simulates measurements of an elastic backscatter lidar.

By using the 3D total concentration as the control variable, the observation operator has the advantage to efficiently assimilate both aerosol optical depth and lidar profiles (separately or jointly). The lidar quantities that could be considered and assimilated within the MOCAGE system are :

- the attenuated backscatter signal
- the aerosol extinction coefficient
- the aerosol backscatter coefficient

**Specific comments :**

1. P5L32 What do you mean about "the increment to be added to the background state is constant over the entire assimilation window"? As my understanding, the assimilation window in your experiment only has one time slot.

➡ **The principle of the 3DFGAT method is illustrated above in Figure R1. The increment resulting from the minimization process is constant over the whole assimilation window independently of its length. In our case, after minimization, the increment is added to the initial state at the beginning of the window. The updated state is then propagated by the model over the assimilation window in order to have the assimilated state on this window.**

**Moreover, the model is propagated over the next window of assimilation to construct the background state which will serve for the assimilation in this window. Below figureR1 illustrates the principle of the 3DFGAT method from _Daget et al., 2009_.**

[Figure]

**Figure R1: A schematic illustration of the 3DFGAT assimilation method. (Courtesy: Daget et al., 2009)**

2. Please also explain the di in the formula 1.

➔ **$d_i = y_i - H_i.x_i^b$ is called the innovation (departure) and represents the distance of the observation $y_i$ from the background $x^b$**
**This is clarified that in the revised version of the paper.**

3. P6L25 and P7L4 Do you mean the two-dimensional diffusion-type equation is the Gaussian function?

➔ **Correct, we clarified this in the revised version of the paper.**

4. P7L6 What are the longitude and latitude lengths? Do you only assimilate only the observations within about 20–22 km? If so, it looks the horizontal lengths are too small. How about the vertical lengths?

➔ **The background covariances, which influence the spread of the analysis to neighboring gridboxes, are specified with constant correlation lengths separately in the horizontal and vertical.**
**The horizontal lengths (longitude and latitude) are constant and are modelled using a Gaussian function (_Pannekoucke and Massart, 2008_) in terms of geographic degrees. In our case, it is fixed to 0.2° in both latitude and longitude. These values are fixed the same as the resolution of the assimilation model. After the assimilation, the modelled state is modified at the observation location and the increment is spread with a Gaussian function around the point of the measurement.**

**The vertical correlation length is modelled, within MOCAGE by using a Gaussian function, in terms of pressure or number of model levels (_Massart et al., 2009_). In this study, the vertical correlation length concerns 2 model levels. The assimilated observation is located in the middle of each layer. After the assimilation, the information is spread in a Gaussian form over the two adjacent layers (the bottom and the top ones).**

5. P7L9 generaly → generally

➔ **Fixed**

6. P7L15 you did not assimilate the AOD in your experiment.

➔ **The AOD observations are not assimilated in our experiment. The purpose of this section is as follows: the total aerosol mass concentration is chosen as the control variable within MOCAGE in order to have an aerosol assimilation system more flexible capable to assimilate AOD observations and lidar profiles (separately or jointly).**
**We have rephrased the sentence to be more general.**

7. P7L17 What do you mean about "the control variable should be the same for all types of observations to be assimilated"?

➔ **This sentence should be situated in a more general context. We have removed this sentence in order to avoid any confusion.**

8. Figure 1, it looks the assimilation system are more effective when the OMF is negative. This probably corresponds to the lower observations and lower observation uncertainties. So the assimilation system give more trust to the observation.

➔ **Correct, moreover it seems that a bias exists between the observations and the background, especially for observations greater than the forecast (positive OMF). The assimilation has reduced this bias especially for this kind of observations.**
**Globally, the assimilation has made the job.**

9. Figure 5, the simulated AODs of both the free run and assimilation experiment are overestimated when the AODs are lower than 0.1. This probably due to the observations from the sites located at high altitude such as >1km.

➔ **Good suggestion. From Figure 5, we remark from the figure that for AERONET AODs lower than 0.1, there is an overestimation of the corresponding values from both the free run simulation and the assimilation. As the reviewer said, it is probably due to observations from high altitude sites.**
**We thank the reviewer for this explanatory element that we added in the new version of the paper.**

10. Figure 8, You compare the simulated aerosol concentrations in the free run and assimilation, however, it is difficult to judge which is better since you do not have the observations. I recommend you compare the simulated extinction coefficients with the CALIOP observations.

➔ **We have added, in the new version manuscript, a new figure comparing CALIOP's observations against the model and against the assimilated product in terms of extinction coefficient. The agreement between the assimilation and CALIOP observations is obvious compared to the comparison between CALIOP measurements and the model free run (Figure R2).**

[Figure]

**Figure R2 : (a) : A measurement orbit of the CALIOP instrument during June 29, 2012 between 12:45 pm and 12:52 pm. (b): The vertical profiles of CALIOP observations in terms of extinction coefficient (m-1). The corresponding profiles from the model free run and the assimilated product is given in (c) and (d), respectively.**

**Response to the comments of the Anonymous Referee #2**

**I - General comments**

This manuscript aims to present, for a case study of a desert dust outbreak in June 2012, the impact of the assimilation of LIDAR observation on a simulation performed with the chemistry transport model MOCAGE. This manuscript has a companion paper (amt-2016-60) focused on the assimilation of aerosol optical depth from MODIS.

Its a topic of scientific interest and within the scope of Atmospheric Measurement Techniques due to the large and rich set of aerosol observations from multiple instruments used to perform the analysis. The presentation of the work is globally very clearly organized, but appears sometimes too limited to a simple superficial description. A more in depth presentation of the results and of some crucial choices would be very useful to improve the added value of this manuscript.

In particular, I have been deeply surprised to find only in the conclusion a mention of a comparison between the previous analysis reported in the companion paper. From my point of view this should be included in the present manuscript.

➔ **The main objective of the paper is to present the lidar profile assimilation system within the MOCAGE model as a complement to the first paper on the AOD assimilation. Thus, the two papers constitute a complete validation of the aerosol assimilation system concerning both AOD and lidar profiles within the MOCAGE system.**
**The paper also addresses a first element concerning the comparison between AOD assimilation and lidar profile assimilation, especially concerning the vertical distribution of aerosols. This was particularly addressed because the approaches concerning the assimilation of the two datasets is totally different: AOD assimilation adopts a uniform correction of the vertical profile of the aerosol concentration over the whole altitude range, whereas lidar profile assimilation is done level by level where the observation is available.**

**It should be noted that both papers deal with a single case study concerning the desert dust transport over few days (10 days). We believe that conclusions concerning the comparison between the assimilation of AOD and lidar profiles based on only one case study will not be conclusive.**
**We believe that for such a comparison to be scientifically conclusive:**
**- It has to be over a long time-period.**
**- It has to concern all types of aerosols and not only desert dust.**
**- It must be done at the global scale to take into account the variability of biases from one region to another (see for example the study conducted by _Shikwambana and Sivakuma, 2018_ which shows that the bias between AOD from CALIOP is not uniform from one region to another at the global scale compared to other instruments: MISR, MODIS,MERRA-2).**
**In addition to all these elements, such a study requires a discussion of the advantages as well as the limitations of each method taking into account the way with which the B matrix is constructed for each type of assimilation including the horizontal and the vertical propagation of the information in the vicinity of the observation location.**

**We believe that addressing all of these elements may take much longer and is beyond the scope of this first validation paper.**

**We should also note that the datasets used for validation in the two papers are not totally identical (product version, number of sites as well as the number of observations per site), which makes a rigorous comparison between the two assimilations a bit complicated.**
**The next step will consist of making a rigorous comparison between the assimilation of AOD and the lidar profiles using the same independent datasets. We will also focus on the advantages as well as the limitations of the assimilation of each type of dataset. We will also consider the possible bias existing between both AOD and lidar products before and after data assimilation.**

On the other hand a large part of the text, providing the general description of the assimilation method (globally not new), the model (not new), the observation database used (not new) which is common with the companion paper could be certainly limited.

**➔ The two papers are published separately. We believe that the reader of each paper should find all the necessary information about the model or the assimilation system in the considered paper without having to go back to the other paper. Nevertheless, we have reduced the parts as recommended by the reviewer.**

**II - Detailed Comments**

The sections 2.2 and 2.3 are largely common with the sections 2 and 3 of the companion paper. They could be largely reduced (or at least put in annex) to focus on the change performed to assimilate LIDAR observation. However, emission used should be mentioned.

**➔ We reduced and merged the two sections as suggested by the reviewer.**
**We have also mentioned the emissions we have used.**

p7, line 1 and 2: The determination of the background error variances and the observation error covariance should be on contrary more detailed. These are results specific to the case study presented.

**➔ See our response to reviewer 1 (question 2)**

p7, line 4-7: This paragraph should be more detailed also. What is the vertical correlation length? How the link between observation with a vertical resolution of 30 to 60m and model with a vertical resolution of 400 to 800 m is made?

**➔ The vertical as well as the horizontal correlation lengths have already been presented in _El Amraoui et al., 2014_. These explanations have been added to the new version of the paper.**

Section 3: I suggest to merge each paragraph of this section with the corresponding one in the section 4 to avoid a dilution effect for the readers.

**➔ We consider that for a coherent structure of the paper, the two sections should remain independent. In fact, section 3 describes all the independent observations**

**used for the evaluation of the model and assimilation products, whereas section 4 presents the results of the presented study.**

section 4.1: Could the definition of observation minus analysis and observation minus forecast be recalled? I understand it is verification of the good behavior of the minimization process. In this case, it could be simply presented like this. Moreover, I think it would be interesting to compare the free forecast and the analysis also in terms of AOD. It could then be more directly compared to the results obtained with the measurements data from MODIS and AERONET.

**➔ Concerning the suggestion of the reviewer to compare the free forecast and the analysis in terms of AOD, It should be recalled that the objective of data assimilation is to evaluate the contribution of observations in a global system compared to the model free run: this is the objective of this paper dealing with the benefit of CALIOP data assimilation compared to MOCAGE outputs.**
**The evaluation of the state of the forecast would have been important if we were in a context of operational forecasting in order to assess the capacity of the model to keep in mind the effect of previous observations. This kind of evaluation is beyond the scope of this paper.**

p12, line 11: "...bias between MOCAGE and MODIS data." I guess?

**➔ Correct, this have been modified in the new version of the paper.**

p13, line 5-6: It is clear that the assimilation of LIDAR data allows a better representation of high values. However, I believe that a comment here on the fact that the positive bias for low values is increased (as shown in figure 5) would be welcome.

**➔ See our response to reviewer 1 (question 9) :**
**Reviewer 1 said that this bias is probably due to observations from high altitude sites. We add this explanatory element in the new version of the paper.**

figure 6 and figure 7: It appears the number concentrations values observed with the ballon flight (∼0.5 cm-3) are much lower than those observed with PCSAP instrument on board the ATR aircraft in the desert dust plume (∼5 cm-3). Any comments on this?

**➔ The size detection ranges of the two instruments are very different. PCASP measures particles over the diameter range 0.1–3 μm, and LOAC over 2-100 μm. We add this element of response in the conclusion.**

figure 7: The horizontal axes could be set to the same scale to ease the comparison between the two flight.

**➔ Fixed**

p8, line 12 : assimilated <-> assimilate

**➔ Fixed**

p8, line 26 : "http://thesesups.ups-tlse.fr/2667/1/2014TOU30293.pdf" <-> Sic (2014)

**➔ Fixed**

p11, line 24 si <-> is

➔ **Fixed**

p34, line 5 the reference to the companion paper should be updated.

➔ **Fixed**

```
* * *
* * *
* * *
```

**References:**

- Cheng, Y., Dai, T., Goto, D., Schutgens, N. A. J., Shi, G., and Nakajima, T.: Investigating the assimilation of CALIPSO global aerosol vertical observations using a four-dimensional ensemble Kalman filter, Atmos. Chem. Phys., 19, 13445–13467, https://doi.org/10.5194/acp-19-13445-2019, 2019.

- Daget, N., A. T. Weaver, et M. A. Balmaseda, 2009a : Ensemble estimation of background-error variances in a three-dimensional variational data assimilation system for the global ocean. Quarterly Journal of the Royal Meteorological Society, 135, 1071-1094.`

- El Amraoui, L., Attié, J.-L., Ricaud, P., Lahoz, W. A., Piacentini, A., Peuch, V.-H., Warner, J. X., Abida, R., Barré, J., and Zbinden, R.: Tropospheric CO vertical profiles deduced from total columns using data assimilation: methodology and validation, Atmos. Meas. Tech., 7, 3035–3057, https://doi.org/10.5194/amt-7-3035-2014, 2014.

- Lahoz et al., (2007), Data assimilation of stratospheric constituents: a review, Atmos. Chem. Phys., 7, 5745–5773

- Massart, S., C. Clerbaux, D. Cariolle, A. Piacentini, S. Turquety, and J. Hadji-Lazaro. First steps towards the assimilation of IASI ozone data into the MOCAGE-PALM system. Atmospheric Chemistry and Physics, 9(14):5073–5091, 2009.

- Ménard and Chang, (2000), Assimilation of Stratospheric Chemical Tracer Observations Using a Kalman Filter. Part II: $\chi2$-Validated Results and Analysis of Variance and Correlation Dynamics. Mon. Wea. Rev., 128, 2672–2686.

- Pannekoucke, O. and S. Massart. Estimation of the local diffusion tensor and normalization for heterogeneous correlation modelling using a diffusion equation. Quarterly Journal of the Royal Meteorological Society, 134(635):1425–1438, 2008.

- Shikwambana, L. and Sivakumar, V.: Global distribution of aerosol optical depth in 2015 using CALIPSO level 3 data, Journal of Atmospheric and Solar-Terrestrial Physics, 173, 150–159, 2018.

- Sič, B., El Amraoui, L., Piacentini, A., Marécal, V., Emili, E., Cariolle, D., Prather, M., and Attié, J.-L.: Aerosol data assimilation in the chemical transport model MOCAGE during the TRAQA/ChArMEx campaign: aerosol optical depth, Atmos. Meas. Tech., 9, 5535–5554, https://doi.org/10.5194/amt-9-5535-2016, 2016.

- Talagrand (2003), A posteriori Validation of Assimilation Algorithms (pp. 85-95), in: R. Swinbank, V. Shutyaev and W. A. Lahoz (editors), Data Assimilation for the Earth System. Dordrecht, The Netherlands: Kluwer Academic Publishers, Nato Science Series.

---

## Author Response (AR2)

July 8, 2020

Laaziz  EL AMRAOUI
Météo-France, CNRM
42, avenue Gaspard CORIOLIS
31057, Toulouse Cedex,
France.
laaziz.elamraoui@meteo.fr

To : Dr. Matthias Beekmann
AMT Editor

Dear Editor,

First of all, we would like to thank you very much for your help during the review process. We would also like to thank the two reviewers for their valuable comments which were very helpful in improving our paper.

On the basis of your last minor revision, we are very happy to send enclosed the last version of the paper with all the comments and corrections you requested (they are noted on boldface).

Thank you again for all the efforts you have made during the review process of this paper.

Yours sincerely,

Laaziz  EL AMRAOUI  on  behalf of all the co-authors